# Intracranial Aneurysm Classifier Using Phenotypic Factors: An International Pooled Analysis

**DOI:** 10.3390/jpm12091410

**Published:** 2022-08-30

**Authors:** Sandrine Morel, Isabel C. Hostettler, Georg R. Spinner, Romain Bourcier, Joanna Pera, Torstein R. Meling, Varinder S. Alg, Henry Houlden, Mark K. Bakker, Femke van’t Hof, Gabriel J. E. Rinkel, Tatiana Foroud, Dongbing Lai, Charles J. Moomaw, Bradford B. Worrall, Jildaz Caroff, Pacôme Constant-dits-Beaufils, Matilde Karakachoff, Antoine Rimbert, Aymeric Rouchaud, Emilia I. Gaal-Paavola, Hanna Kaukovalta, Riku Kivisaari, Aki Laakso, Behnam Rezai Jahromi, Riikka Tulamo, Christoph M. Friedrich, Jerome Dauvillier, Sven Hirsch, Nathalie Isidor, Zolt Kulcsàr, Karl O. Lövblad, Olivier Martin, Paolo Machi, Vitor Mendes Pereira, Daniel Rüfenacht, Karl Schaller, Sabine Schilling, Agnieszka Slowik, Juha E. Jaaskelainen, Mikael von und zu Fraunberg, Jordi Jiménez-Conde, Elisa Cuadrado-Godia, Carolina Soriano-Tárraga, Iona Y. Millwood, Robin G. Walters, Helen Kim, Richard Redon, Nerissa U. Ko, Guy A. Rouleau, Antti Lindgren, Mika Niemelä, Hubert Desal, Daniel Woo, Joseph P. Broderick, David J. Werring, Ynte M. Ruigrok, Philippe Bijlenga

**Affiliations:** 1Neurosurgery Division, Department of Clinical Neurosciences, Faculty of Medicine, Geneva University Hospitals, 1205 Geneva, Switzerland; 2Department of Pathology and Immunology, Faculty of Medicine, University of Geneva, 1205 Geneva, Switzerland; 3Stroke Research Centre, University College London Queen Square Institute of Neurology, London WC1N 3BG, UK; 4Department of Neurosurgery, Canton Hospital St. Gallen, 9000 St. Gallen, Switzerland; 5ZHAW School of Life Sciences and Facility Management, 8820 Wädenswil, Switzerland; 6Institut National de la Santé et de la Recherche Médicale (INSERM), Centre National de la Recherche Scientifique (CNRS), University Hospital Centre Nantes, University of Nantes, L’institut Du Thorax, 44007 Nantes, France; 7Department of Neuroradiology, University Hospital of Nantes, 44000 Nantes, France; 8Department of Neurology, Faculty of Medicine, Jagiellonian University Medical College, ul. Botaniczna 3, 31-503 Krakow, Poland; 9Neurogenetics Laboratory, The National Hospital of Neurology and Neurosurgery, London WC1N 3BG, UK; 10Department of Neurology and Neurosurgery, University Medical Center Utrecht Brain Center, Utrecht University, 3584 CS Utrecht, The Netherlands; 11Department of Medical and Molecular Genetics, Indiana University School of Medicine, Indianapolis, IN 46202, USA; 12Department of Neurology and Rehabilitation Medicine, University of Cincinnati College of Medicine, Cincinnati, OH 45267, USA; 13Departments of Neurology and Public Health Sciences, University of Virginia School of Medicine, Charlottesville, VA 22903, USA; 14Department of Interventional Neuroradiology—NEURI Brain Vascular Center, Bicêtre Hospital, APHP, 94270 Le Kremlin Bicêtre, France; 15Institut national de la santé et de la recherche médicale (INSERM), CIC 1413, Clinique des Données, University Hospital Centre Nantes, 44000 Nantes, France; 16Department of Neuroradiology, Dupuytren University Hospital, 87000 Limoges, France; 17Department of Neurosurgery, Helsinki University Hospital, University of Helsinki, 00260 Helsinki, Finland; 18Clinical Neurosciences, University of Helsinki, Topeliuksenkatu 5, 00260 Helsinki, Finland; 19Neurosurgery Research Group, Biomedicum, 00290 Helsinki, Finland; 20Department of Vascular Surgery, Helsinki University Hospital, University of Helsinki, 00290 Helsinki, Finland; 21Department of Computer Science, University of Applied Science and Arts, 44139 Dortmund, Germany; 22Institute for Medical Informatics, Biometry and Epidemiology (IMIBE), University Hospital Essen, 45147 Essen, Germany; 23SIB Swiss Institute of Bioinformatics, 1015 Lausanne, Switzerland; 24Diagnostic and Interventional, Department of Diagnostics, Faculty of Medicine, Geneva University Hospitals, 1205 Geneva, Switzerland; 25Division of Neurosurgery, Department of Surgery, St Michael’s Hospital, University of Toronto, Toronto, ON M5B 1W8, Canada; 26Swiss Neuroradiology Institute, 8002 Zurich, Switzerland; 27Lucerne School of Business, Lucerne University of Applied Sciences, 6002 Lucerne, Switzerland; 28Neurosurgery NeuroCenter Kuopio, University Hospital Kuopio, 70210 Kuopio, Finland; 29Institute of Clinical Medicine, Faculty of Health Sciences, University of Eastern Finland, 70210 Kuopio, Finland; 30Institut Hospital del Mar d’Investigacions Biomèdiques (IMIM) and Hospital del Mar, 08003 Barcelona, Spain; 31Clinical Trial Service Unit and Epidemiological Studies Unit, Nuffield Department of Population Health, University of Oxford, Oxford OX1 2JD, UK; 32MRC Population Health Research Unit, University of Oxford, Oxford OX1 2JD, UK; 33Department of Anesthesia and Perioperative Care, Center for Cerebrovascular Research, University of California, San Francisco, CA 94143, USA; 34Department of Epidemiology and Biostatistics, Institute for Human Genetics, University of California, San Francisco, CA 94143, USA; 35Department of Neurology, University of California, San Francisco, CA 94143, USA; 36Montreal Neurological Institute and Hospital, McGill University, Montréal, QC H3A 0G4, Canada; 37Department of Clinical Radiology, Kuopio University Hospital, 70210 Kuopio, Finland

**Keywords:** intracranial aneurysm, subarachnoid hemorrhage, risk factors, location, smoking, hypertension

## Abstract

Intracranial aneurysms (IAs) are usually asymptomatic with a low risk of rupture, but consequences of aneurysmal subarachnoid hemorrhage (aSAH) are severe. Identifying IAs at risk of rupture has important clinical and socio-economic consequences. The goal of this study was to assess the effect of patient and IA characteristics on the likelihood of IA being diagnosed incidentally versus ruptured. Patients were recruited at 21 international centers. Seven phenotypic patient characteristics and three IA characteristics were recorded. The analyzed cohort included 7992 patients. Multivariate analysis demonstrated that: (1) IA location is the strongest factor associated with IA rupture status at diagnosis; (2) Risk factor awareness (hypertension, smoking) increases the likelihood of being diagnosed with unruptured IA; (3) Patients with ruptured IAs in high-risk locations tend to be older, and their IAs are smaller; (4) Smokers with ruptured IAs tend to be younger, and their IAs are larger; (5) Female patients with ruptured IAs tend to be older, and their IAs are smaller; (6) IA size and age at rupture correlate. The assessment of associations regarding patient and IA characteristics with IA rupture allows us to refine IA disease models and provide data to develop risk instruments for clinicians to support personalized decision-making.

## 1. Introduction

Approximately 3% of the population harbor an unruptured intracranial aneurysm (IA), and the overall risk of aneurysmal rupture is about 1% per year [1]. Rupture of IA causes aneurysmal subarachnoid hemorrhage (aSAH), and despite recent improvements in risk stratification, imaging, surgical techniques, and intensive care treatment, functional outcome after aSAH remains poor [1]. As ruptures often occur at a young age with high case fatality (~50%), aSAH substantially reduces productive life-years [2]. Additionally, up to 76% of patients who survive a hemorrhage have permanent cognitive deficits and remain dependent; only 6–17% return to work [3]. Predicting aneurysm rupture allows for the possibility of potentially preventing aSAH, an important clinical and socioeconomic goal.

Risk factors potentially associated with IA rupture include patient characteristics (i.e., family history of IA, previous history of aSAH, hypertension, and hypercholesterolemia), IA characteristics (i.e., aneurysm location, aneurysm size, and number of aneurysm), patient habits (i.e., smoking, and use of aspirin) and demographic characteristics (i.e., age, female sex, and ethnicity) [4,5,6,7,8,9,10,11,12,13,14]. Although predictive models such as the PHASES score [6] have been created to identify patients that would benefit from intervention, their clinical use is limited. More refined modelling of the effects of patient and IA characteristics on IA rupture is needed. As longitudinal studies that assess the risks of IA rupture are inevitably biased by case selection, are very resource intensive, and require multiple years of observation, we decided to use retrospective data available to the International Stroke Genetics Consortium–Intracranial Aneurysm Group (ISGC-IA) to perform an initial cross-sectional analysis that may be followed by a complementary longitudinal study. For the present study, our aim was to investigate differences in patient and IA characteristics that are routinely assessed in clinical practice between patients diagnosed with unruptured IA versus patients diagnosed in the context of aSAH, and to provide a multivariate logistic regression model to quantify the magnitude and strength of association of the studied factors with the likelihood of being diagnosed with an unruptured versus ruptured IA.

## 2. Materials and Methods

### 2.1. Patients

Data collected from several genetic studies on IA [15] were shared among the International Stroke Genetics Consortium–Intracranial Aneurysm Group (ISGC-IA) (Appendix A and Appendix A). Detailed cohort descriptions are given in Appendix A. For all participating groups, approval was received by their institutional or regional ethics committee [15]. Written informed consent was obtained for all participating patients.

### 2.2. Data Collection, Patient and IA Characteristics

Initial assessment of the available data allowed identification of a minimal data set of seven phenotypic patient characteristics (basis of recruitment (diagnosis of incidental IA, symptomatic IA, or aSAH), sex, family history of IA, hypertension, smoking, age at time of SAH, IA multiplicity) and three aneurysm characteristics (rupture status, maximum diameter at rupture, IA location) that could be harmonized across all studies (Appendix A).

### 2.3. Data Sources

We determined the distributions of sex, hypertension status and smoking status in reference populations for comparison with the patients in our total cohort (*n* = 8560). According to United Nations data for 2010 (data.un.org export on 4th November 2018), the proportion of females in the general population was 50.9%. In a reference population generated using data from the European Commission website [16] (extracted on 4th November 2018 for the year 2010 in relevant countries), the CoLaus Study [17], and other relevant sources [18,19,20,21]), 19.4% self-reported high blood pressure (HBP), and HBP was prevalent in 50.9%. Smoking data in a reference population generated using data from the European Commission website [16], the OCDE website [22] (extracted on 4th November 2018 for 2010), the CoLaus Study [17], and other relevant sources [23,24]) revealed that 19.8% were current smokers, 31.8% were former smokers, and 48.4% were non-smokers.

### 2.4. Data Analysis

We performed a cross-sectional analysis of patient and IA characteristics associated with IA rupture status at diagnosis. As age and size of IA at diagnosis in patients diagnosed with unruptured aneurysm were considered of limited biological relevance, these two parameters were not included in the analysis.

In Figure 1 and Figure 2, the assessment of differences in the distribution of cases by categories of factors between groups is illustrated using mosaic plots in which the box size is proportional to the number of people in the corresponding cells. Groups were compared using Pearson Chi-square tests. Pearson residuals describe the overall agreement between observed and fitted or expected values; they approximately follow a normal distribution, which implies that two-sided *p*-values of 0.05, 0.01, 0.001, and 0.0001 correspond approximately to standardized residuals of 2.0, 2.6, 3.3, and 3.9, respectively. In Figure 3, the odds ratio (OR) and relative risk of aneurysm rupture for each location category were calculated using the MCA location as the reference category. The ORs were based on median-unbiased estimates, and the 95% confident interval (CI) was determined using the mid-p method [25]. In Figure 4 and Figure 5, data are shown as a boxplot, with median values shown below each plot. The differences in the distribution of cases by size and age at rupture between groups were assessed using Wilcoxon tests. The threshold of statistical significance was set at *p* < 0.001 and Bonferroni correction applied for multiple comparison assessing seven hypotheses. In Figure 6, the association between IA rupture status at diagnosis and factors was analyzed using multivariable logistic regression (MLoR). Association of factors with IA size and age at rupture was analyzed using multivariate linear regressions (MLiR). Results were reported using OR and 95% confidence interval (CI). For the MLiR model, results were reported as factor estimates and 95% CI. Receiver operating characteristics (ROC) curves were used to assess the performance of the MLoR. An area under the curve (AUC) of 0.5, >0.7, >0.8, >0.9, and 1 indicates no, acceptable, good, excellent, and perfect discriminative ability, respectively. Analysis was conducted in R version 3.6.1. (R Core Team, R Foundation for Statistical Computing, Vienna, Austria; https://www.r-project.org).

### 2.5. Data Available Statement

The raw dataset is archived at the Aneurysm Data Bank [26]. It can be made available upon request for auditing purposes or further research. Specific requests for access to aggregated data will be granted after formal acceptance by the consortium. Access to patient-level data will be subject to adherence to the European Data Protection Regulation (GDPR 2016/679). R code will be shared upon request from any qualified investigator.

## 3. Results

### 3.1. Characteristics of Whole Cohort

A total of 8560 patients were recruited among the various contributing studies. Data regarding IA diagnosis (i.e., incidental, symptomatic, or aSAH) were missing or discordant in 53 patients (0.6%). In addition, 94 (1.1%) had missing data for IA rupture status at diagnosis, and 421 (4.9%) had missing data for IA location. Therefore, the analyzed cohort included 7992 (93.4%) patients (Appendix A).

Females were over-represented (68.5%) compared with the reference population (50.9%) (Figure 1A). Median age at aneurysm rupture was 52 years (range 10–92). Data on family history were available for 91.4% of patients (Figure 1B). Information on aneurysm multiplicity was available for 98.1% (Figure 1C). Patients with multiple aneurysms had from 2 to 10 lesions (Figure 1D). Information on blood pressure, available for 97.2%, was compared with an equal sample size of self-reported and expected prevalence of high blood pressure (HBP) in the reference population (Figure 1E). Smoking status was available in 92.2% (Figure 1F). In the analyzed cohort, 44.3%, 23.3%, and 32.4% of the participants were current smokers, former smokers, and non-smokers, respectively. This result represents a significantly higher proportion of smokers compared with the reference population (19.8%, 31.8%, and 48.4%, respectively) (Figure 1F). Aneurysms located in the Acom (26.7%), MCA (25.9%), and Pcom (14.7%) account for two-thirds of the total number of aneurysms in the analyzed cohort.

### 3.2. Patient Characteristics and Likelihood of Being Diagnosed with aSAH

Patients in the analyzed cohort were diagnosed more often with aSAH than with unruptured IAs (68%; Appendix A). In contrast, patients with a positive family history were diagnosed more often with unruptured IAs (OR (95%CI): 2.2 (2.0–2.4), *p* < 0.001) than sporadic cases, most likely due to IA screening. However, 52.7% of the patients with a positive family history of IA were diagnosed only after IA rupture (Figure 2A). Sixty percent of patients had normal blood pressure (NBP) or were not aware of HBP. Patients with known HBP were more likely diagnosed with unruptured IA (1.4 (1.3–1.6), *p* < 0.001) (Figure 2B). Current smokers were more likely diagnosed with aSAH (1.5 (1.3–1.6), *p* < 0.001) and former smokers with incidental IAs (2.1 (1.9–2.3), *p* < 0.001). Non-smokers were more frequently diagnosed with aSAH (1.2 (1.1–1.4), *p* < 0.001) (Figure 2C). The proportion of patients with unruptured and ruptured IA at diagnosis were similar in Finnish compared to North American and/or European patients, in male and female, as well as in patients with solitary and multiple IAs (Figure 2D–F).

### 3.3. IA Location and Risk of Rupture

We arbitrarily defined IA locations with ruptures in more than 75% of the patients as high risk for rupture, and we defined locations with ruptures in less than 25% of the patients as low risk for rupture. As illustrated in Figure 3A,B, PCA, Acom, A2, Pcom, and VB locations may be classified as high risk for rupture, and Ophtl-ICA and Cav-ICA locations may be classified as low risk for rupture. MCA, ICA, Basilar, A1 and other locations may be considered as medium risk.

### 3.4. IA Location and Size at Rupture

Figure 3C shows the distribution of IA size at rupture for the three location subgroups based on risk. Median IA size at rupture was significantly smaller in the high-risk locations (6 mm (IQR 4–8)) compared with the medium-risk (7 mm (5–10), *p* < 0.001) and low-risk locations (8.5 mm (6–13), *p* < 0.001). The sample size was insufficient to determine a significant difference between the medium-risk and low-risk locations. One percent of all ruptured IAs were smaller than 2 mm, 6.5% were <4 mm, and 16% were <7 mm. In high-risk locations, 22% of IAs were <4 mm, and 54% were <7 mm. In medium-risk locations, 17% were <4 mm and 41% were <7 mm, and in low-risk locations, 12% were <4 mm and 27% were <7 mm.

### 3.5. IA Location and Age at Rupture

Figure 3D shows the distribution of age at rupture for the low-, medium-, and high-risk IA location groups. Mean age at rupture was significantly higher in high-risk locations (53 years (IQR 44–62)) compared with medium-risk (51 years (43–60), *p* < 0.001). The mean age at rupture of low-risk locations was younger although not reaching our significance threshold level (47 years (41–54), *p* = 0.0014). No relevant difference in age distribution at rupture was found between medium- and low-risk locations.

### 3.6. Patient Age at IA Rupture

Age at IA rupture was available for 5419 patients. Median age at aneurysm rupture was older for females (53 (IQR 45–62) years) compared with males (50 (42–58), *p* < 0.001) (Figure 4A). Median age at rupture was younger for patients with a positive family history of IA (49 (41–59) years) compared with those with sporadic aneurysms (52 (44–61) years, *p* < 0.001) (Figure 4B). The age difference between patients with solitary and multiple aneurysms (52 (44–61) and 51.5 (44–61) years, respectively) was not significant (Figure 4C). Median age at rupture was older for patients with known HBP (56 (47–65) years) compared with those who did not self-report HPB (i.e., had NPB or were unaware of HBP) (50 (42–58) years, *p* < 0.001) (Figure 4D), and it was younger for current smokers (50 (42–58) years) compared with former smokers (54 (47–63) years, *p* < 0.001) and non-smokers (55 (45–64) years, *p* < 0.001) (Figure 4E). The age difference between former smokers and non-smokers was not significant.

### 3.7. IA Size and Risk of Rupture

Aneurysm diameter at rupture was available for 4091 patients (75%). Maximum diameters ranged from 0.2 mm to 68 mm, and median size at rupture was 6.0 mm (4.0–9.0). Differences in size between females (6.0 mm (4.0–8.0)) and males (6.0 mm (4.5–9.0), *p* < 0.01) did not meet our threshold for statistical significance (Figure 5A). Differences in size between patients with a positive family history of IA (6.8 mm (5.0–9.0)) and patients with sporadic IA (6.0 mm (4.0–8.0), *p* < 0.01) also did not meet our threshold (Figure 5B). Median size at rupture was smaller for patients with solitary aneurysm (6.0 mm (4.0–8)) compared with those with multiple IAs (7.0 mm (5.0–9.0), *p* < 0.001) (Figure 5C). IA size at rupture did not differ between patients with known HBP or unknown HBP (both 6.0 mm (4.0–9.0)) (Figure 5D). IA median size at rupture was larger for current smokers (6.0 mm (5.0–9.0)) compared with non-smokers (5.5 mm (4.0–8.0), *p* < 0.001). The comparison between current smokers and former smokers (6.0 mm (4.0–9.0), *p* < 0.01) did not meet our threshold for significance (Figure 5E). The difference between former and non-smokers was not significant.

### 3.8. Classifiers

Multiple logistic regression revealed that IA location, smoking status, and awareness of HBP were the factors with the highest association with rupture status (Figure 6A). Being aware of having HBP or being a former smoker was associated with a higher likelihood of unruptured IA status at diagnosis. IAs located in high- or medium-risk locations, being aware of HBP, or being female was associated with IA rupture at an older age, whereas being a current smoker or having Finnish background was associated with IA rupture at a younger age (Figure 6B). Other factors appeared to have either no effect or only a small effect on age at IA rupture. IA located in high- or medium-risk locations or IA in females ruptured at smaller sizes (Figure 6C). IA in smokers (current and former) ruptured at larger sizes (Figure 6C). IA multiplicity and positive family history did not seem to have an influence on patient age or IA size at rupture. ROC curve showing performances of the MLoR is shown in Figure 6D. AUC was 0.73 (95% CI: 0.70–0.76), indicating an acceptable discriminative ability between patients diagnosed with unruptured IA and aSAH.

## 4. Discussion

Our cross-sectional analysis of aneurysm status at diagnosis showed that IA location is the principal factor associated with IA rupture. Awareness regarding HBP and smoking cessation increase the likelihood of incidental diagnosis with unruptured IA. Females, despite being more likely to be diagnosed with IA than males, are proportionally as likely as males to be diagnosed with ruptured IAs. Our observations regarding the effect of IA location on IA rupture status at diagnosis is comparable to previous reports [11,27,28].

In addition, IA size at rupture is strongly associated with IA location: IAs in high-risk locations are smaller upon rupture compared with medium- and low-risk locations. Only 1% of ruptured IAs were <2 mm regardless of the location. Interestingly, approximately one out of four aneurysms ruptured at <7 mm in low- (27%), <5 mm in medium- (26%), and <4 mm in high-risk locations (22%). This observation suggests that an IA size 4–7 mm might be critical depending on the IA location. This finding is in agreement with previous longitudinal studies such as ISUIA, UCAS, and PHASES [5,6,8].

Several hypotheses that explain differences in disease initiation and evolution at different locations exist. Blood flow and associated wall shear stress seem to be important factors that are highly influenced by angio-architecture and bifurcation shapes [29,30,31,32]. Vessel wall structure, influenced by differences in angio- and vasculo-genesis, as well as inflammation or perivascular environment may also play a role [33,34]. As it is likely that small unruptured asymptomatic IAs are underdiagnosed [35] and IA size can alter when they rupture, the probability of rupture depending on IA size was not evaluated.

Our study highlighted that being aware of risk factors increases the likelihood of being diagnosed with an unruptured IA rather than aSAH. Indeed, awareness of HBP was more frequently associated with unruptured IA. Diagnosis of IA rupture in patients with HBP occurs at an older age. No association between HBP awareness and aneurysm size at rupture was seen.

Smoking is considered as one of the strongest and most consistent risk factors for aneurysm formation and rupture [36,37,38]. In our cohort, the proportion of current smokers was higher in patients diagnosed with IA, compared with the reference population. This confirms the effect of active smoking on disease initiation. Moreover, the proportion of current smokers was higher in patients diagnosed with aSAH compared with patients diagnosed with unruptured IA; this suggests an effect of smoking on disease evolution as well. Current smokers tend to have smaller aneurysms at rupture, indicating early rupture in these patients [9]. The proportion of former smokers in the IA cohort was smaller than the proportion in the reference population. In addition, former smokers were less frequently diagnosed with aSAH compared with unruptured IA. This might indicate that smoking cessation leads to a decrease in risk of IA rupture. This again is in line with most previous studies [36]. The fact that non-smokers had a similar likelihood of developing aSAH compared with the combined group of current and former smokers suggests that smoking reversibly activates the disease. Therefore, recommendation for current smokers diagnosed with an IA to stop smoking is crucial.

In accordance with previous studies, a positive family history of IA increases the likelihood of being screened and consequently diagnosed with an unruptured IA. Therefore, positive family history is a protective factor for aneurysm rupture as it raises disease awareness, triggers early intervention and increases detection of modifiable factors such as hypertension. However, it is important to note that in subjects with a positive family history who would have warranted screening, most IAs were diagnosed only after aSAH. Moreover, we cannot exclude that patients with a positive family history might have died prematurely and would consequently not have been included in the present study, resulting in a bias in the number of patients having a positive family history for IA and affected by aSAH. Such an observation suggests room for improvement with regard to aneurysm screening.

Although IA multiplicity is a marker of IA susceptibility, IA multiplicity was not associated with IA rupture, IA size at rupture, or patient age at rupture. These observations support the hypothesis that disease initiation and evolution do not share the same underlying pathophysiological mechanisms. Patients with multiple IAs are more prone to aneurysm initiation, but their aneurysms might have a risk of rupture similar to that of solitary IAs, and the overall risk is determined by the lesion located at the highest-risk location.

In line with previous studies [39,40,41], women are overrepresented in our cohort of patients affected by IAs. However, the proportions of females in the ruptured and unruptured IA subgroups are similar, which suggests that females are more likely to develop IAs but their aneurysms do not rupture more frequently. Some studies report no difference between males and females < 50 years of age, but incidence of unruptured IA and aSAH increases in postmenopausal women, which suggests a hormonal element [42,43]. Interestingly, females in our cohort had a higher median age at rupture compared with males. Therefore, although females have a higher prevalence of unruptured IA, their aneurysms might be more stable. We previously showed in a small cohort that endothelial cell coverage of the intraluminal surface of IA domes was higher in females [44], which could add to a higher stability of the IA dome.

Our study has several limitations. It is a cross-sectional study that compares the status of patients and aneurysms at time of diagnosis and therefore needs to be interpreted with caution regarding the effects of factors on aneurysm evolution and risk of rupture that can only be formally assessed using a longitudinal study design. Nevertheless, a significant fraction of the cohort of patients was recruited prospectively and consecutively on a population basis, reducing case-selection bias (*n* = 1164, 14.6%). Case-selection bias is a major limitation of longitudinal studies as cases identified at risk of rupture are mostly treated and excluded. In addition, patients with severe aSAH are more likely to die and are thus less likely to be included in studies, resulting in overoptimistic models. We therefore propose to combine observations from both cross-sectional and longitudinal studies into an integrated disease model. The longitudinal observations would be reported as soon as sufficient follow-up is available on that sub-cohort of cases.

Basis of recruitment differed among the various sub-cohorts. Although some sub-cohorts recruited their patients prospectively and consecutively, recruitment was mainly retrospective, and some sub-cohorts limited their study to patients who contributed DNA samples. Nevertheless, when we performed the same analyses limited to the group of prospectively, consecutively recruited population-based sub-cohorts, we were able to replicate all reported findings. However, the CIs were larger and power to detect statistical significance was reduced.

Retrospective harmonization of data may introduce some mapping problems and missing data. Mapping IA locations was unequivocal with the exception of ICA and ophtl-ICA, which were not separate location categories in one cohort that focused only on aSAH. This may have contributed to underestimating the relative risk of rupture of ICA aneurysms. However, excluding this study from analysis resulted in a relative risk that remained within the CI range. Another harmonization limitation is that exact blood pressure measurements were recorded only for a small subgroup of patients. Most sub-cohorts recorded diagnosis of HBP as yes/no responses from questionnaires, and only a few collected data on blood pressure control by medication, limiting the possibility to accurately assess the impact of HBP on IA rupture. It is important that blood pressure is systematically evaluated in future prospective studies.

Our study also has considerable strengths. We present the largest cohort to date of patients with ruptured and unruptured IAs. Special care in the methodology has been taken to reduce the impact of potential bias. We have limited our analysis to factors that could be robustly harmonized across different data sources, and we confirmed the consistency of our results among different sub-cohorts. Finally, our statistical significance threshold was set conservatively to reduce the risk of false-positive signals. Despite some limitations, this study defines and contextualizes IA patients who will be recruited and followed up in longitudinal studies to address the highly relevant question of which unruptured IA goes on to rupture and has to be treated.

## 5. Conclusions

This cross-sectional analysis of patient and IA characteristics produced a MLoR classifier and two MLiR models that quantify the magnitude and strength of associations between the studied factors and IA rupture status at diagnosis, as well as the association of factors with IA size and patients’ age at rupture. It establishes locations in the brain that are most strongly associated with rupture risk and finds that the association of IA location with HBP awareness and smoking habit are the most relevant factors to estimate the odds of IA rupture. We showed that IA location, HBP awareness, sex, smoking habit, and ethnic background have significant associations with age at IA rupture, and that IA location, sex, and smoking habit are associated with IA size at rupture. These factors as well as genetics, IA geometry and environment, blood viscosity, cerebrovascular autoregulation, and vessel wall composition should be assessed more precisely and followed-up over time to refine the IA disease model and provide risk assessment instruments that support personalized decision-making. We do believe that the classifier proposed in this study can be used in longitudinal patient cohorts in order to develop a model to identify unruptured IAs with a substantial rupture risk.

## Figures and Tables

**Figure 1 jpm-12-01410-f001:**
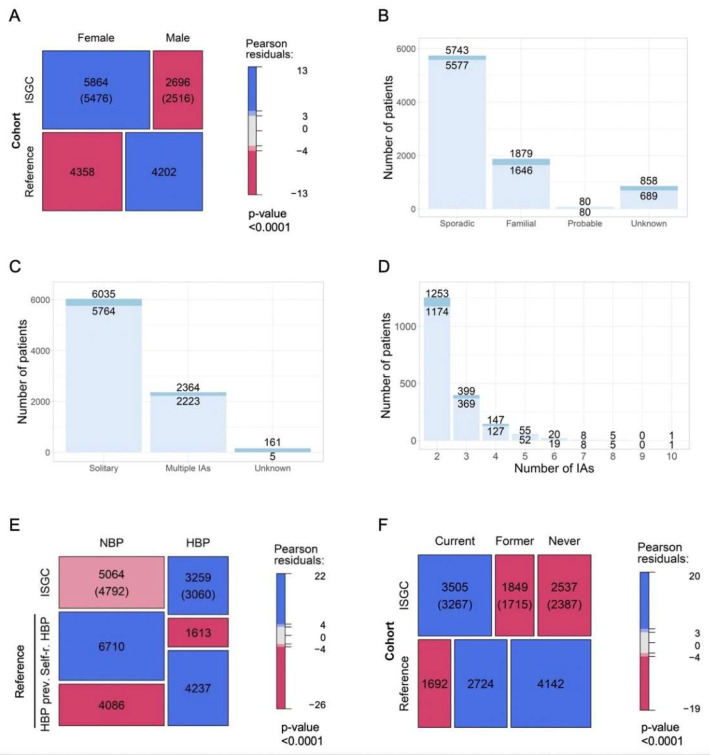
Cohort characteristics. Distribution of patients by sex (**A**), family history of IA (**B**), IA number (**C**,**D**), blood pressure (**E**), and smoking status (**F**). In panels (**B**–**D**), light blue represents participants with known IA location, and dark blue represents participants with missing or conflicting information. In panels (**A**,**E**,**F**), counts in parentheses correspond to the number of patients enrolled with known IA location. IA: Intracranial aneurysm; NBP: Normal Blood Pressure. Self-r. HBP: self-reported High Blood Pressure in the reference population, HBP prev.: HBP prevalence.

**Figure 2 jpm-12-01410-f002:**
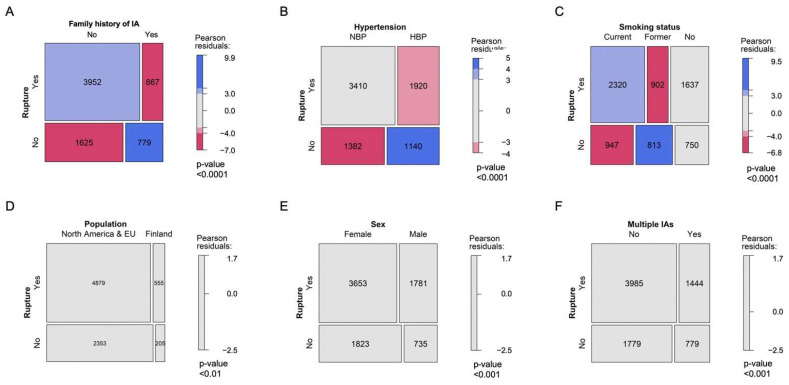
Patient characteristics and diagnosis of aSAH. Effect of family history of IA (**A**), blood pressure (**B**), smoking status (**C**), patient geographic location (**D**), sex (**E**), and IA multiplicity (**F**) on IA rupture status at diagnosis.

**Figure 3 jpm-12-01410-f003:**
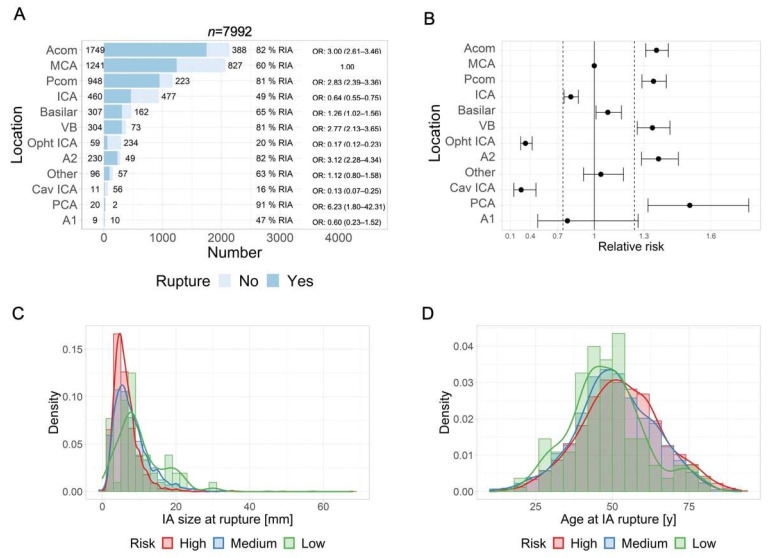
Intracranial aneurysm locations and risk of rupture. (**A**) Number of ruptured (dark blue) and unruptured (light blue) aneurysms for each IA location. (**B**) Relative risk for IA rupture, by IA location. The solid vertical line represents the relative risk for MCA. The dashed lines separate low-risk locations (left side) and high-risk locations (right side) from medium-risk locations (middle). (**C**) Distribution of IA size at rupture in high-risk (Acom, Pcom, VB, A2, and PCA, in red), medium-risk (MCA, ICA, basilar, A1, and other locations, in blue) and low-risk (Opht-ICA and Cav-ICA, in green) locations for rupture. (**D**) Distribution of patient’s age at rupture in high-risk (red), medium-risk (blue), and low-risk (green) IA locations for rupture. Acom: anterior communicating artery; MCA: middle cerebral artery; Pcom: posterior communicating artery; ICA: internal carotid artery; VB: vertebra-basilar artery, ophtl-ICA: ophthalmic segment of ICA, A2: anterior cerebral artery distal to Acom, cav-ICA: carotid-cavernous ICA, PCA: other posterior circulation arteries, A1: A1 anterior segment.

**Figure 4 jpm-12-01410-f004:**
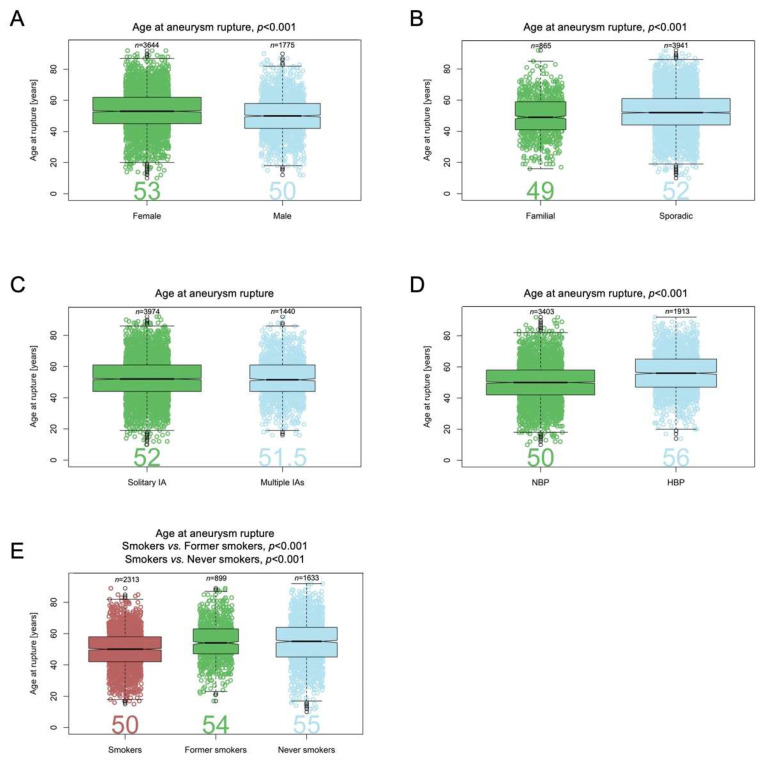
Patient’s age at time of rupture and risk factors for rupture. Patient’s age at IA rupture by sex (**A**), family history of IA (**B**), IA multiplicity (**C**), blood pressure (**D**), and smoking status (**E**).

**Figure 5 jpm-12-01410-f005:**
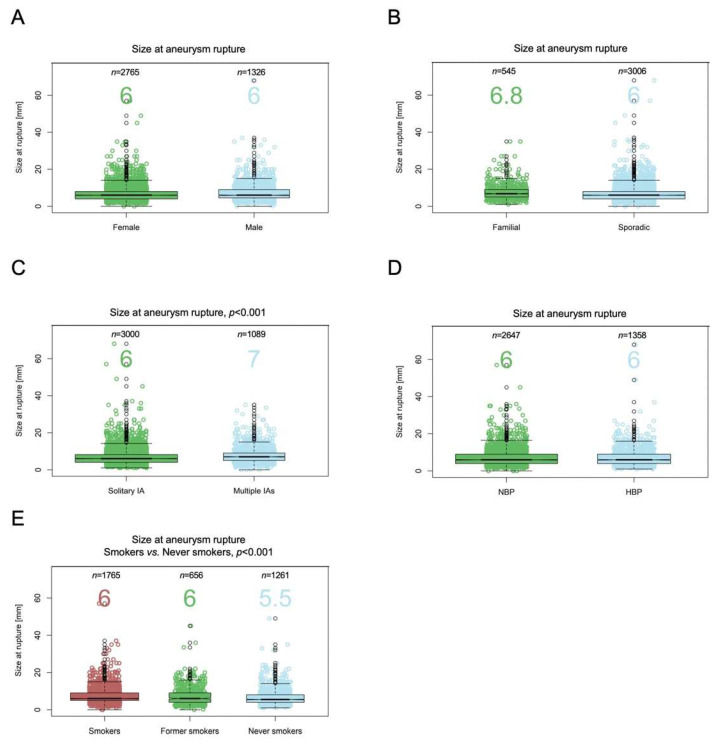
Intracranial aneurysm size at rupture and risk factors for rupture. IA size at rupture by sex (**A**), family history of IA (**B**), IA multiplicity (**C**), blood pressure (**D**), and smoking status (**E**).

**Figure 6 jpm-12-01410-f006:**
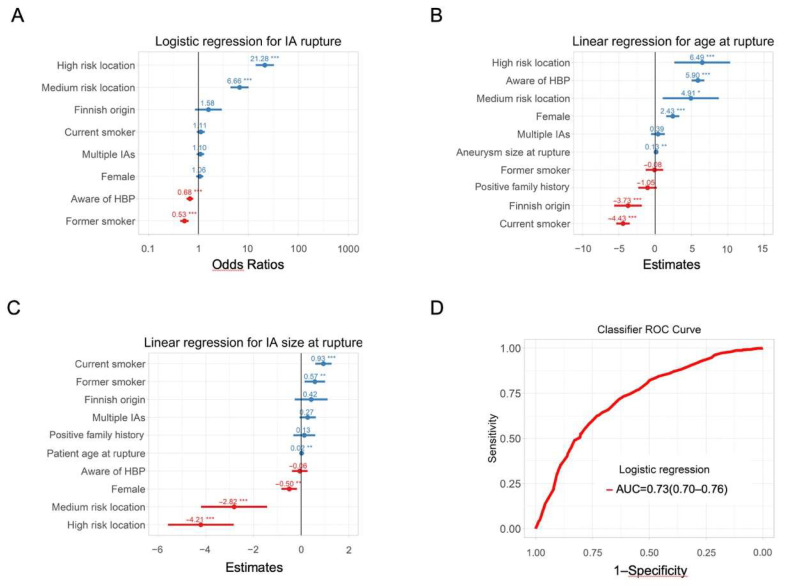
Importance of phenotypic markers for the likelihood of being diagnosed with a ruptured IA. (**A**) MLoR for diagnosis of aSAH. (**B**) MLiR for patient’s age at rupture and (**C**) MLiR for IA size at rupture. * *p* < 0.05, ***p* < 0.01, *** *p* < 0.001. (**D**) ROC curve of the MLoR.

## Data Availability

The raw dataset is archived at the Aneurysm Data Bank [26]. It can be made available upon request for auditing purposes or further research. Specific requests for access to aggregated data will be granted after formal acceptance by the consortium. Access to patient level data will be subject to adherence to the European Data Protection Regulation (GDPR 2016/679). R code will be shared upon request from any qualified investigator.

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
