# Peer review of "Intracranial Aneurysm Classifier Using Phenotypic Factors: An International Pooled Analysis"

_jpm, 2022, doi:10.3390/jpm12091410_

Round 1

Reviewer 1 Report

The authors analyzed about 8,000 patients with intracranial aneurysms (IAs) from International Stroke Genetics Consortium–Intracranial Aneurysm Group. They demonstrated association of some patient characteristics (smoking, IAs location, etc) with the likelihood being diagnosed with an unruptured or ruptured IA. I completely agree with the author's opinion that more refined modelling of the effects of patient and IA characteristics on IA rupture is needed.  Unfortunately, they did not developed a model for identifying unruptured IAs at risk of rupture in the present study. I think it the critical limitation.

In addition, there are several issues that need to be addressed for publication.

1. They investigated the difference between patient characteristics with unruptured IAs and ones with ruptured IAs. Because the present report is not an observational study, it cannot show risk factors for aneurysmal rupture but can just show the likelihood being diagnosed with an unruptured or ruptured IA.

It seems inappropriate to classify IA location into 3 categories (high-, medium-, low-risk of rupture) and refer to relative risk for IA rupture by IA locations using ratio of unruptured and ruptured IAs for each IA location in figure 3.

2. For the same reason, they should avoid use of 3 categories of IA location regarding risk of aneurysmal rupture in figure 6.

3. In figure 6, presentation of ROC curve and AUC gives the false impression that it contributes to Identifying unruptured IAs at risk of rupture. Therefore, I think it should be removed.

Reviewer 2 Report

Authors have conducted a thorough, sound, and extensive labor of research. The authors sought to perform a cross-sectional analysis of patients with intracranial aneurysms. Their findings support the actual body of knowledge and although their results are not quite innovative their methods are solid, and the investigation has been beautifully conducted and reported.

Author Response

We thank the reviewer for her/his analysis of our manuscript.

We are deeply touched by her/his acknowledgment of the extensive labor of research that support the actual body of knowledge and is based on a solid methodology and investigation.

We would like to express our gratitude to the reviewers and editor giving us the opportunity to address their questions and remarks.

Best regards

Sandrine Morel and Philippe Bijlenga on behalf of the ISGC – IA consortium

Reviewer 3 Report

Very good work. English language and style are fine/minor spell check required

Author Response

We thank the reviewer for the positive feedback. We have gone through the manuscript and checked again for spelling mistakes. All changes are track changed.

Sandrine Morel and Philippe Bijlenga on behalf of the ISGC – IA consortium

Reviewer 4 Report

See attached file.

Author Response

We thank the reviewer for her/his careful evaluation of our manuscript. Our answers are attached in the file named "Revision-Reviewer 4".docx".

Round 2

Reviewer 1 Report

Unfortunately, the resubmitted paper has not been modified based on our requests. The likelihood being diagnosed with ruptured IA is not the same as rupture risk of IAs. The use of the word, “rupture risk” or presentation of ROC curve and AUC gives the false impression that it contributes to the prediction for unruptured IAs at risk of rupture. As the authors mentioned, recruitment of different cohorts is an advantage in the present study. However, inclusion criteria are not consistent across cohorts. For example, some cohort include only patients with ruptured IAs, other cohort include ones with ruptured and unruptured IAs. I meant observational longitudinal study in the former comment. This is not the essence of the argument. This cross-sectional study did not provide an accurate risk assessment of IA rupture.

Author Response

We thank the reviewer for the detailed evaluation of our manuscript. Based on the comments of the reviewer provided during the first and the second round of the revision, we have improved our manuscript following her/his remarks.

To not give the false impression that our study quantifies the IA rupture risk, we have modified the text accordingly:

  • Title 3.2. “Patient characteristics and IA risk of rupture” changed to “Patient characteristics and likelihood of being diagnosed with aSAH” (see track changes, page 7, line 245).
  • Title figure 2: “Patient characteristics and IA risk of rupture” changed to “Patient characteristics and diagnosis of aSAH” (see track changes, page 8, line 260).
  • Title figure 6: “Importance of phenotypic markers for IA risk of rupture” changed to “Importance of phenotypic markers for the likelihood of being diagnosed with a ruptured IA” (see track changes, page 13, lines 350-360).
  • We have added some more clarifying information:
    • “Despite some limitations, this study defines and contextualises IA subpopulation of patients who will be recruited and followed up in longitudinal studies to address the highly relevant question of which unruptured IA goes on to rupture and has to be treated” (page 16, lines 469-472).
    • “We do believe that the classifier proposed in this study can be used in longitudinal patient cohorts in order to develop a model to identify unruptured IAs with a substantial rupture risk” (page 16, lines 486-488).

We hope that we addressed the remarks and were able to convince the reviewer.

Sandrine Morel and Philippe Bijlenga on behalf of the ISGC – IA consortium

Round 3

Reviewer 1 Report

No further comment